# Feasibility Study of an Educational Intervention to Improve Water Intake in Adolescent Soccer Players: A Two-Arm, Non-Randomized Controlled Cluster Trial

**DOI:** 10.3390/ijerph18031339

**Published:** 2021-02-02

**Authors:** Rubén Martín-Payo, María del Mar Fernández-Álvarez, Edurne Zabaleta-del-Olmo, Rebeca García-García, Xana González-Méndez, Sergio Carrasco-Santos

**Affiliations:** 1Faculty of Medicine and Health Sciences, University of Oviedo, Campus del Cristo s/n, 33006 Oviedo, Spain; martinruben@uniovi.es; 2PRECAM Research Team, Health Research Institute of the Principality of Asturias, Avenida Roma s/n, 33011 Oviedo, Spain; quecaoviedo@gmail.com (R.G.-G.); xxanina@hotmail.com (X.G.-M.); sergio.carrasco@sespa.es (S.C.-S.); 3Fundació Institut Universitari per a la recerca a l’Atenció Primària de Salut Jordi Gol i Gurina (IDIAPJGol), 08007 Barcelona, Spain; ezabaleta@idiapjgol.org; 4Gerència Territorial de Barcelona, Institut Català de la Salut, 08007 Barcelona, Spain; 5Nursing Department, Nursing Faculty, Universitat de Girona, 17004 Girona, Spain; 6Campus Bellaterra, Universitat Autònoma de Barcelona, 08193 Barcelona, Spain; 7Hospital Universitario Central de Asturias, Avenida Roma s/n, 33011 Oviedo, Spain; 8Hospital Universitario San Agustín de Avilés, Camino de Heros 6, 33401 Avilés, Spain

**Keywords:** soccer, behavior, health promotion, adolescent, body hydration status

## Abstract

This study aimed to assess the feasibility of an educational intervention on hydration behavior in adolescent soccer players. A pilot study of a two-arm, non-randomized controlled cluster trial was conducted. A total of 316 players aged 13–16 agreed to participate. The response variables were the players’ participation in the intervention, their perception of the knowledge acquired, the usefulness and the overall assessment of the intervention. Hydration patterns and acquisition of knowledge on hydration behavior were also assessed. The intervention involved two elements: posters and a web app. A total of 259 adolescents completed the study (intervention group (IG) = 131; control group (CG) = 128). 80.6% of the players responded to the survey assessing the feasibility of the intervention. The mean number of correct answers regarding behavior was significantly higher in the IG (3.54; SD = 1.162) than in the CG (2.64; SD = 1.174) (*p* < 0.001). The water consumption pattern at all the clubs was ad libitum. Of the players, 10% did not drink any water at all during the game. In conclusion, this intervention has been shown to be feasible for implementation with adolescent soccer players. It suggests that hydration guidelines should be informed by personal factors and that ad libitum water consumption should be avoided.

## 1. Introduction

Soccer, one of the most frequently played sports in the world [1], requires players to have a high physical capacity and to develop technical and tactical skills [2], resulting in high demand for water during play [1].

The water needs of individuals are influenced by multiple personal factors, such as anthropometric characteristics, sex, and age [3]. In the case of athletes, specific needs may arise depending on the characteristics of the physical activity [4,5] or the environmental conditions present [2,4,5,6], which generally result in increased water consumption. In this context, a water deficit can negatively affect aerobic and cognitive performance, the development of strength and power, and the player’s technical qualities [7,8,9,10].

Although numerous studies have assessed interventions involving underage populations, most of them focus on physical activity and eating behaviors [11,12,13] and on the school environment [14,15,16]. By contrast, despite the fact that soccer is a widely known and commonly played sport, very few studies have been conducted on educational interventions related to hydration behavior in children or adolescents.

For these interventions to be effective, they must be adapted to the target context and population [15,17]. For example, it is recommended that interventions involving minors include creative and fun activities [16] and use digital resources, as these tools are widely accepted by underage populations [18]. In addition, it is of paramount importance that the design and implementation of educational interventions be informed by theoretical frameworks. In this regard, the Behavior Change Wheel (BCW) model, developed by Michie et al. [19], is particularly useful. Despite being a novel tool, it has already proven to be effective in designing and assessing behavior change interventions with minors [20,21,22,23].

Given the huge importance of appropriate water intake in this context, this study aimed to assess the feasibility of an intervention focusing on hydration behavior in soccer players aged 13 to 16 in the Principality of Asturias (Spain) based on the BCW model.

## 2. Materials and Methods

### 2.1. Design

This is a pilot study of a two-arm, non-randomized controlled cluster trial of an educational intervention designed to improve the water intake of adolescent soccer players during play.

### 2.2. Participants

The study was conducted with soccer players aged 13 to 16 from clubs in the Principality of Asturias (Spain) during the 2018–2019 season. Given the study design, the intervention was implemented in two clubs (2 intervention clubs or IGs), to which other clubs with similar characteristics were subsequently added until a number of players similar to those of the intervention clubs was reached (3 control clubs or CGs).

All players whose parent or parents authorized their participation by signing and returning the informed consent form were included. The following exclusion criteria were considered: non-completion or inadequate completion of the questionnaires by the players or their legal guardians, failure to attend the training session during which the data were collected and request for withdrawal from the study by the players themselves or their legal guardian.

A total of 423 invitations were sent, and 316 players volunteered to participate: 161 in the IGs and 155 in the CGs. A cluster allocation design was used. As a result, players from the same club were all part of the same group, thus avoiding cross-contamination among the participants.

### 2.3. Measurement Instruments and Study Variables

In order to meet the study objectives, the following response variables were used: the players’ participation in the intervention, their perception of the knowledge acquired, the perceived usefulness of the intervention, and the overall assessment of the intervention. Hydration patterns, water intake, and acquisition of knowledge on hydration behavior were also assessed.

To assess the feasibility of the intervention, players from the intervention clubs completed a self-administered questionnaire designed ad hoc. In this questionnaire, participants were asked to make an overall assessment of the educational intervention using response options ranging from 0 (lowest score) to 10 (highest score). They were also asked to express their perception of the knowledge they had acquired on hydration and of the usefulness of the educational intervention using dichotomous response options (“I have/I have not acquired knowledge” and “The intervention was useful/not useful”). The following aspects were also calculated: the players’ adherence rate to the intervention (expressed as the percentage of adolescents who agreed to participate in the study and completed the assessments), the percentage of players who consulted the educational posters related to hydration behavior (“consulted the posters: yes/no”) and the percentage of players who accessed the web app (“accessed the web app: yes/no”). Finally, the mean number of times the web app was accessed per player was calculated. These variables were assessed during a training session.

The water consumption pattern established by each club to ensure underage players’ hydration was ascertained by asking the coaches for each team. The water consumption variable was expressed as the amount of water, in milliliters (mL), consumed by the players during an official match. This variable was measured by a researcher (MFA) pre-intervention and post-intervention, who asked players about the amount of water they consumed. Additionally, given the influence of environmental conditions and physical effort on water intake, the following were also measured: the number of minutes played, the environmental temperature, and the relative humidity during the game. The mean environmental temperature and relative humidity were measured at the beginning and end of the match using the OREGON SCIENTIFIC BAR-206 weather station. Data collection was conducted simultaneously at all clubs before the intervention, in September–October 2018, and after the intervention, in May–June 2019, during an official match.

At the end of the educational intervention, all players were surveyed using a self-administered questionnaire designed ad hoc to assess the knowledge acquired. This questionnaire included five questions relating to hydration behavior with four possible response options, only one of which was correct. The questions answered correctly added 1 point each. Unanswered questions and questions answered incorrectly neither added nor subtracted points. The final score was the total sum of correct responses (range: 0 = less knowledge; 5 = more knowledge).

### 2.4. Intervention

The educational intervention lasted 6 months and was based on the BCW model. The backbone of this model is known as COM-B, which stands for capability, opportunity, motivation, and behavior [19] (Table 1). The players in the control clubs did not receive any kind of intervention. Instead, they continued with their usual training and competition activities. In accordance with expert recommendations [24], in the design phase, factors that could influence the success of the intervention, such as the location of the intervention, the resources available, or the information channels preferred by the participants, were analyzed. Given the context and the experiences reported in previous studies [15,16,25], the use of a web app and posters displayed in places accessible to the minors within the sports grounds were chosen as tools which, a priori, would ensure higher levels of participation and greater effectiveness.

The collaboration and involvement of sports club managers and coaches was also solicited in order to create a social environment conducive to change and provide appropriate conditions for implementing the intervention. This educational intervention, designed specifically for the players, included two key elements: posters and a web app.

The posters featured content related to hydration behavior, e.g., factors influencing proper hydration, the amount of fluid needed before and after competing, symptoms of dehydration, types of drinks, etc. The posters were displayed in strategic locations at the sports facilities (locker rooms, corridors, bulletin board) and were changed on a weekly basis. Coaches were entrusted with the task of persuading the participants to read the posters in order to create a social environment conducive to change.

The web app, supervised by a nutrition technician, was designed and developed specifically for this study. The hydration-related information it contained was updated weekly (e.g., the water content of different foods, the benefits of drinking water for the body, rehydration strategies, recommended beverages, etc.) In addition, forms with questions regarding the content of both the posters and the web app were included in the web app on a bi-monthly basis. The aim of these forms was to motivate players to read the information on both the web app and the posters.

### 2.5. Ethical Approval

All players participated voluntarily. Consent was obtained in writing from the players’ legal guardians to participate in this study. The study was conducted according to the guidelines of the Declaration of Helsinki. In addition, permission was requested from the Research Ethics Committee of the Principality of Asturias (project number: 59/2018), and the Spanish Public Prosecutor’s Office for Minors was informed.

### 2.6. Statistical Analysis

Measures of central tendency and dispersion were calculated for quantitative variables. Absolute values and percentages were calculated for categorical variables. Fulfillment of the normality criterion was checked using the Kolmogorov–Smirnov test. Between-groups and within-groups comparisons of the study variables were also performed. Categorical variables were compared using either Pearson’s chi-squared test or McNemar’s test. Quantitative variables were compared using non-parametric tests (Mann–Whitney U test and Wilcoxon’s signed-rank tests) for non-normal distributions. The associations between environmental variables and water intake and between physical activity time and water intake were explored using Spearman’s rank correlation coefficient. The analyses were performed using the IBM^®^ software SPSS^®^ Statistics, version 24.0.0.0. The statistical significance threshold for all tests was set at 0.05.

## 3. Results

### 3.1. Description of the Sample and the Environmental Conditions

Of the 316 players selected, 18.03% (*n* = 57; 33 in the CG and 24 in the IG) were considered to be lost to follow-up because they failed to attend the training session during which the water consumption data were collected. As a result, 259 adolescents (mean age = 14.2 years; SD = 1.078) completed the study (128 from the CG and 131 from the IG) (Figure 1).

The pre-intervention median duration of physical activity was 92.0 min (IQR = 40.00), and the post-intervention median duration of physical activity was 98.0 min (SD = 26.513). The pre-intervention median temperature was 7.5 degrees Celsius (IQR = 9.00), and the pre-intervention relative humidity was 76.5% (IQR = 16.00). The post-intervention median temperature was 14.0 degrees Celsius (IQR = 11.00) and the post-intervention relative humidity was 68.0% (IQR = 9.00). No significant pre-intervention differences between the groups with respect to temperature and time of physical activity were found. However, the number of minutes of physical activity was slightly higher in the IG both before and after the intervention (Table 2).

### 3.2. Feasibility of the Intervention

Of the 155 players in the IG, 131 participated in the activities, and 80.6% (*n* = 125) responded to the survey assessing the feasibility of the intervention. The participants rated the intervention with a mean score of 7.9 points (SD = 1.574); 83.1% of the participants considered that the intervention had improved their knowledge of hydration behavior, while 88.5% of the participants believed that the intervention had been useful. The posters were seen by 100% of the participants, and 68% of them accessed the web app. The web app registered 409 visits, representing a mean of 4.81 visits per player.

### 3.3. Water Consumption and Knowledge Acquisition

None of the participating clubs had established a consumption pattern whereby minors were advised on the amount of water they should consume. The pattern used by 100% of the clubs was ad libitum, that is, at the discretion of the minors.

After the educational intervention, knowledge related to hydration behavior was measured by surveying players in both the IG and the CG. The mean number of correct responses attained by the participants in the IG was 3.54 (SD = 1.162), while those in the CG obtained 2.64 (SD = 1.174). This difference was found to be statistically significant (*p* < 0.001).

The median amount of water consumed before the intervention was 150.0 mL (IQR = 250.00), and the median amount of water consumed after the intervention was 125.0 mL (IQR = 250.00). No significant differences between the groups were observed. The level of water consumption decreased between the pre- and post-intervention phases, as shown in Table 3.

Approximately 10% of the minors reported not drinking any water during the game (10.3% before the intervention and 10.8% after the intervention). These figures did not reveal any significant differences between the groups (Table 3).

Finally, correlation analyses were performed to explore the association between environmental variables (temperature and humidity) and water intake, and between the duration of physical activity (minutes) and water intake, both before and after the intervention, with no correlation between them being observed (Table 4).

## 4. Discussion

The results of this study demonstrate the feasibility of an innovative educational intervention based on the use of a web app in conjunction with other tools. The intervention was well accepted by adolescent soccer players and improved their knowledge of hydration but did not result in increased water intake. The amount of water consumed by players was lower than that observed in similar studies [6,26,27,28], and 10% of the participants did not drink any water at all during the course of a game. This is rather alarming, as the amount of water consumed may be insufficient to meet minors’ needs during sports activities.

The literature indicates that interventions incorporating contextual modifications are more effective [29]. However, given that this was a pilot study to assess feasibility, the intervention was designed to meet the minimum requirements only. As a result, it focused on enhancing players’ performance by improving their knowledge of hydration without addressing other organizational or environmental aspects, such as providing access to water or modifying the hydration patterns established by each team.

The posters were seen by 100% of the players. The strategic location of the posters and the actions of the coaches to persuade participants to read them may have contributed to creating a social environment conducive to change [30]. Moreover, the results showed a more than adequate acceptance of the web app as an educational tool among participants, demonstrating the impact of this technology on knowledge acquisition. However, the number of minors who accessed the website showed room for improvement. It should be noted that the players who accessed the web app as well as seeing the posters obtained a slightly higher number of correct responses. As indicated in the theoretical model employed, knowledge acquisition is directly related to improving psychological capacity and reflective motivation, which are key determinants in developing healthy behaviors [30]. An increase in the frequency of use of the web app is likely to result in significantly higher scores in the knowledge variable.

The water consumption pattern at all participating clubs was ad libitum. This pattern was not modified in the educational intervention, which may help explain the lack of increase in water consumption. In this regard, Duffield et al. [31] and Owen et al. [32] suggest avoiding ad libitum consumption and establishing water consumption patterns as an effective measure to prevent dehydration. Several studies conducted in sports contexts reinforce this idea by demonstrating that an ad libitum water consumption pattern is not efficient to cover the need for fluids, particularly in children and adolescents [33,34,35]. By way of example, in a study on soccer players by Laitano et al. [2], it was observed that the fluids ingested replaced approximately 50% of the fluids lost when an ad libitum pattern was followed.

There appears to be a general trend towards low fluid intake in minors both in Spain [36] and in other countries [37,38]. This can be detrimental to health and is further aggravated when minors engage in physical exercise. Therefore, when planning seasons, competitions, and training sessions, it is advisable to include hydration and water repletion strategies tailored to players’ personal needs. It is essential that minors drink fluids during sport as they offer numerous benefits, such as improved sports performance [8,9] and prevention of dehydration.

Diagnosing dehydration in an individual may prove difficult due to the large number of factors to be taken into consideration, such as environmental conditions [2,5,6,33] and the duration and intensity of exercise [4,26]. In this study, the state of hydration was not assessed. Dehydration could occur despite moderate weather conditions because weather conditions are indirectly related to dehydration. However, no significant association was observed between water intake and environmental conditions. Instead, an association was observed between water intake and increased physical activity, i.e., the most active children consumed more water. This could be because their bodies were alerting them to the need to hydrate. Therefore, there is a real risk of dehydration while playing sports. It is essential to analyze the extent to which an individual may be at risk of dehydration and implement strategies to prevent it in all age groups by avoiding the associated risks and favoring their normal growth and development [39,40].

Finally, the acquisition of knowledge on hydration observed among players in the intervention clubs is another successful aspect of the educational intervention [41]. As indicated in the theoretical model employed, knowledge acquisition is directly related to improving psychological capacity and reflective motivation, which are key determinants in developing healthy behaviors [19].

One of the limitations of the present study is the non-inclusion of female players. The results of this study may therefore be considered applicable to male players only. It was not possible to analyze the reasons for the percentage of use of the web app. Future research should include female players and establish mechanisms to analyze the causes of non-use of the web app and other tools included in the intervention. Our study was designed for a specific population and future studies should also adapt their interventions to each target population. The research was conducted in an area where the climate is not extreme, so the result could differ in hot/cold areas. It could be interesting to replicate the research in extreme weather areas. Finally, the strength of our study lies in highlighting the need to improve water consumption. Future study designs could include modifications of an environmental and/or organizational nature.

## 5. Conclusions

The intervention, based on the BCW model, has been shown to be feasible for implementation in adolescent soccer players aged 13 to 16. The results of this study may therefore be considered applicable to male players only. It was not possible to analyze the reasons for the percentage of use of the web app. Future research should include female players and establish mechanisms to analyze the causes of non-use of the web app and other tools included in the intervention.

In practice, hydration guidelines should be based on personal factors, ad libitum water consumption should be avoided, and environmental and/or organizational modifications may be included.

## Figures and Tables

**Figure 1 ijerph-18-01339-f001:**
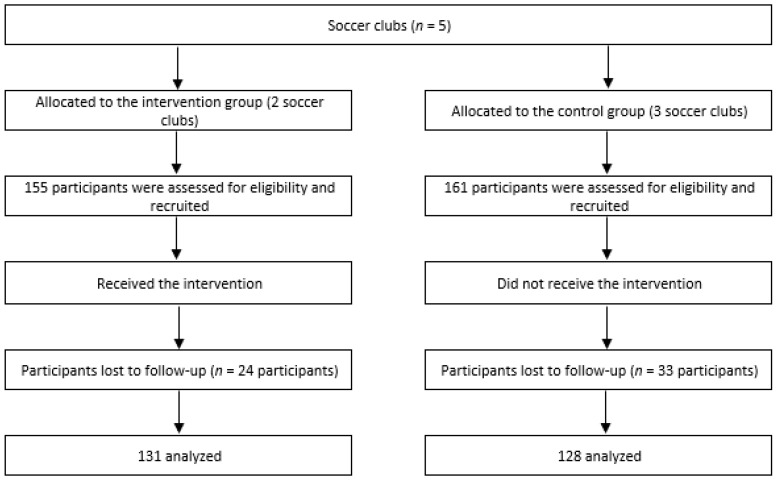
Flow diagram of participants.

**Table 1 ijerph-18-01339-t001:** Determinants, intervention functions, and needs included in the intervention.

COM-B	Needs Addressed by the Intervention	Intervention Functions
Psychological capability	Providing knowledge related to hydration and water repletion	Education
Physical capability	Developing the skills required to follow hydration guidelines	Entertainment
Social opportunity	Generating a positive culture among players and coaches for optimal, healthy hydration	Enablement Social influence
Physical opportunity	Providing players with the means to implement and develop hydration guidelines and routines	Environmental restructuring
Reflective motivation	Assessing the health and performance benefits of proper hydration and following appropriate hydration guidelines	Education Persuasion Incentivization
Automatic motivation	Increasing players’ confidence and promoting a sense of belonging	Persuasion

**Table 2 ijerph-18-01339-t002:** Pre- and post-intervention median values for temperature, humidity, and physical activity time for the CG (*n* = 128) and the IG (*n* = 131), and the differences between the groups.

Variables	Pre/Post Intervention	CG Median (IQR)	IG Median (IQR)	*p **
Temperature	Pre	12.5 (7.50)	7.5 (9.50)	0.126
	Post	14.0 (4.00)	14.0 (4.50)	0.140
Humidity	Pre	73.5 (20.00)	79.5 (14.50)	0.080
	Post	65.5 (4.50)	71.5 (23.5)	0.008
Physical activity time (minutes)	Pre	92.0 (41.00)	87.5 (43.25)	0.024
	Post	100.0 (37.50)	95.0 (43.50)	0.004

* Mann–Whitney nonparametric tests.

**Table 3 ijerph-18-01339-t003:** Pre- and post-intervention median values for water intake and knowledge for the CG (*n* = 128) and the IG (*n* = 131), and the differences between the groups.

Variables	Pre/Post Intervention	CG	IG	*p*
Water intake (mL)	Pre	150.0 (325.00)	150.0 (212.50)	0.712 *
Median (IQR)	Post	137.5 (250.00)	125.0 (225.00)	0.611 *
*p*		0.144 *	0.768 *	
% of players who did not drink water during the game	Pre	12.6	8.1	0.198 ^†^
	Post	10.9	10.7	0.948 ^†^
*p*		0.824 ^‡^	0.648 ^‡^	
Knowledge (number of correct responses; range: (0–5)—Median (IQR)		3.0 (2.00)	4.0 (2.00)	<0.001 *

* Nonparametric tests (Mann–Whitney U test or Wilcoxon’s signed-rank test); ^†^ Pearson’s chi-squared test; ^‡^ McNemar’s test.

**Table 4 ijerph-18-01339-t004:** Spearman’s rank correlation coefficients between water intake, environmental conditions, and physical activity time.

Variables	Pre-Intervention Water Intake (mL)	Post-Intervention Water Intake (mL)
Environmental temperature (degrees Celsius)	0.161 *	0.145 *
Environmental humidity (%)	−0.192 ^†^	0.105
Physical activity time (minutes)	0.229 ^†^	0.241 ^†^

* *p* ≤ 0.05; ^†^
*p* < 0.001.

## Data Availability

The data presented in this study are available on request from the corresponding author. The data are not publicly available due to the ethical concern.

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
