# Peer review of "Feasibility Study of an Educational Intervention to Improve Water Intake in Adolescent Soccer Players: A Two-Arm, Non-Randomized Controlled Cluster Trial"

_ijerph, 2021, doi:10.3390/ijerph18031339_

Round 1

Reviewer 1 Report

This paper is well written and has a novel approach to hydration practices.  I do have some suggestions and one concern.

Suggestions

1.  In the abstract - you need to define IG and CG (line 27) - it is not defined until the reader gets to Methods section

2.  It was not clear who actually measured the water consumption of the players - this is critical to your study and needed to validate the amount of water the athletes consumed (lines 104-1190

Concern - the measured median temp at the Pre (7.5 IG and 12.5 CG) and Post (14.0 C for both groups) is quite cold and therefore not a good example of hydration consumption during HOT/Extreme weather conditions.  This is a concern since hydration needs to be high during warmer temps (140 C/57.20 F) is rather cold and not a suitable temperature to raise core temp that would necessitate increased water consumption.

Author Response

In the abstract - you need to define IG and CG (line 27) - it is not defined until the reader gets to Methods section

Thank you very much for pointing this out. We defined IG and CG in the abstract.

It was not clear who measured the water consumption of the players - this is critical to your study and needed to validate the amount of water the athletes consumed (lines 104-1190

Thank you very much for your comment. Water consumption was measured by the researcher MFA.

Concern - the measured median temp at the Pre (7.5 IG and 12.5 CG) and Post (14.0 C for both groups) is quite cold and therefore not a good example of hydration consumption during HOT/Extreme weather conditions.  This is a concern since hydration needs to be high during warmer temps (140 C/57.20 F) is rather cold and not a suitable temperature to raise core temp that would necessitate increased water consumption.

Thank for your comment. We completely agree with the reviewer. The literature indicates that environmental temperature acts as predictor of water consumption. Nonetheless, we felt that our findings fulfilled the aim of our study, which was “to assess the feasibility of an intervention focusing on hydration behavior”. Future studies could be conducted in warmer regions to verify the feasibility of the intervention.

Reviewer 2 Report

Dear authors,

This is a well conducted and well written piece of work on hydration. The age group selected and the context is applicable to many types of physical activity.

It is succint, reads well and the design appeared to be well thought of.The population was well defined and there was a clear intervention and control.

Check out fig 1, although it started as control and intervention, on the right hand, it ended as intervention again i.e 128 participan completed the intervention (control is not intervention), this needs to be changed. 

It is not clear from the method how water consumption during an official match was measured e.g were players asked to record it in a diary, did officials keep tab, was it self-reported to researchers during questioning etc 

The authors made an attempt to explain the differences between the two groups and how cross-contamination was avoided.

It was good to see that environmental condition was measured and reported alongside water intake as a potential cofounder. 

It is a feasilbility study and I hope enough has been learnt to design a larger trial. 

Author Response

Check out fig 1, although it started as control and intervention, on the right hand, it ended as intervention again i.e 128 participan completed the intervention (control is not intervention), this needs to be changed.

Thank you for pointing this out. The term was changed in Figure 1.

It is not clear from the method how water consumption during an official match was measured e.g were players asked to record it in a diary, did officials keep tab, was it self-reported to researchers during questioning etc

Thank you. We have included the method in the text: “asked players about the amount of water they consume.”

The authors made an attempt to explain the differences between the two groups and how cross-contamination was avoided. It was good to see that environmental condition was measured and reported alongside water intake as a potential cofounder.

Thank you very much for your kind words. We pay special attention to methodological aspects throughout the project.

Reviewer 3 Report

Thank you for the opportunity to review this article. The work is interesting, but some aspects should be taken into account before publication.

Comments and suggestions for Authors:

Introduction:

  • Authors should include a literature review in the introduction.
  • Authors should improve the introduction including the latest articles published for example in the MDPI platform or others, about other research in this field
  • Authors should better justify the study, highlighting clearly the gap in the current knowledge. What is the novelty of the study?

Materials and Methods

  • Please discuss power calculation and how the sample size is adequate.
  • Why authors include exactly 316 subjects? How many invitations did you send? Please write more about recruitment of the study group.
  • Have authors validated the questionnaire?

Discussion:

Limitations of this study should be noted. Strengths of this study should be noted.

Author Response

Introduction:

Authors should include a literature review in the introduction.

Thank you for your recommendation. We failed to find a literature review suitable for the introduction. However, we believe that the references included in that section have a high methodological quality and suits the purposes of the Introduction section.

Authors should improve the introduction including the latest articles published for example in the MDPI platform or others, about another research in this field

A new reference (41) was included in the Discussion section.

Authors should better justify the study, highlighting clearly the gap in the current knowledge. What is the novelty of the study?

Adolescents who play sports are still at risk of dehydration. For instance, as seen in our results, 10% of the soccer players did not drink any water during the game. This justifies the need to develop interventions to improve hydration in this population. Furthermore, we were unable to find sufficient research-supported evidence in Spain.

Materials and Methods

Please discuss power calculation and how the sample size is adequate.

Our study was educational in nature, designed to be a pilot study for application to a specific population, which is why we did not use a sampling method, but rather worked with the entire population available. However, we included this as a limitation in the Limitations section.

Why authors include exactly 316 subjects? How many invitations did you send? Please write more about recruitment of the study group.

Thank you for your questions. We have added this in the manuscript.

Have authors validated the questionnaire?

The questionnaire used is educational in nature. However, prior to its use, it was independently validated by 8 experts in the fields covered by the questionnaire.

Discussion:

Limitations of this study should be noted. Strengths of this study should be noted.

This section was added at the end of the Discussion section.

Round 2

Reviewer 1 Report

I still feel that a limitation to this study is due to the environmental conditions during which the study took place.  Temps between 7 - 140 C are quite cool and as a result a true representation of an individual's hydration practices may not be adequately observed when compared to environmental conditions where environmental temps are above 280 C and hydration practice is more of a concern.

Author Response

Thank for the comment. As suggest the reviewer a limitation was added in the text "The research was conducted in an area where the climate is not extreme, so the result could differ in hot / cold areas. It could be interesting to replicate the research in extreme weather areas".